# Potential Use of Gut Microbiota Composition as a Biomarker of Heat Stress in Monogastric Species: A Review

**DOI:** 10.3390/ani11061833

**Published:** 2021-06-19

**Authors:** Yuqing He, Christian Maltecca, Francesco Tiezzi

**Affiliations:** Department of Animal Science, North Carolina State University, Raleigh, NC 27695, USA; yhe22@ncsu.edu (Y.H.); cmaltec@ncsu.edu (C.M.)

**Keywords:** elevated ambient temperatures, heat stress, gut microbiota, monogastric animal, biomarker, health, welfare

## Abstract

**Simple Summary:**

Heat stress is a significant environmental challenge faced by food animal production worldwide because of its adverse effects on animal performance and productivity. Trillions of microorganisms living in the gut are essential for host health by participating in various digestive, immune, and metabolic activities. At the same time, they are known to be sensitive to changes in the surrounding environment. The present review summarizes current research progress of how the gut microbial community responds to elevated ambient heat in monogastric animal species and discusses the use of the gut microbiota composition as a potential indicator for heat stress.

**Abstract:**

Heat stress is a current challenge for livestock production, and its impact could dramatically increase if global temperatures continue to climb. Exposure of agricultural animals to high ambient temperatures and humidity would lead to substantial economic losses because it compromises animal performance, productivity, health, and welfare. The gut microbiota plays essential roles in nutrient absorption, energy balance, and immune defenses through profound symbiotic interactions with the host. The homeostasis of those diverse gut microorganisms is critical for the host’s overall health and welfare status and also is sensitive to environmental stressors, like heat stress, reflected in altered composition and functionality. This article aims to summarize the research progress on the interactions between heat stress and gut microbiome and discuss the potential use of the gut microbiota composition as a biomarker of heat stress in monogastric animal species. A comprehensive understanding of the gut microbiota’s role in responding to or regulating physiological activities induced by heat stress would contribute to developing mitigation strategies.

## 1. Introduction

Meat and dairy provided by food animal production are essential sources of nutrition to feed the growing human population worldwide. According to the Food and Agricultural Organization (FAO), food animal production respectively provides more than 340, 840, and 80 million tons of meat, dairy, and egg products worldwide each year [1]. The swine and poultry industries are essential components of agricultural production and are significant sources of animal proteins.

Heat stress (HS) is a severe environmental challenge for agricultural animal production. According to the Intergovernmental Panel on Climate Change (IPCC) report, global temperatures have increased about 1 °C from the pre-industrial period to 2017 [2]. They are predicted to continue rising by about 0.2 °C per decade, which, if left unchecked, may surpass the limits set by the Paris Agreement to protect the whole natural and human systems from detrimental heat stress [2,3]. Furthermore, in recent years, substantial heat extremes with prolonged hot days and more frequent hot waves have been observed in many regions or countries, particularly in Africa, Southeast Asia, central and south America [4], and Europe [5]. Thus, the impact of heat stress on agricultural animals is exacerbated by the continued rise in global temperatures and the increased frequency of extreme heat events.

When the ambient temperature approaches thermoneutrality for most homeotherms, the body reduces metabolic processes that generate heat and increases heat dissipation through conduction, convection, and radiation mechanisms [6]. Once the ambient temperature exceeds thermoneutrality, physiological activities, including sweating, increased respiration and panting, become the primary routes for heat dissipation [6]. However, some animal species like pigs and birds lack sweat glands for heat evaporation, making them more sensitive to the increase in ambient temperature [7,8]. Besides, intense genetic selection for improved productivity also makes animals more susceptible to heat stress due to increased metabolic heat loads generated for fast growth [9]. When the elevated ambient temperatures increase the core body temperature and induce a series of changes in physiological activities and behaviors, the animal is under heat stress [10].

Trillions of diverse microorganisms colonizing the gastrointestinal (GI) tract (referred to as ‘microbiota’) are extensively involved in nutrient absorption, energy balance, and immune defenses through profound symbiotic interactions with the host [11]. A healthy and stable intestinal microbial community has direct influences on host health and performance. Oppositely, the disturbance in intestinal microbiota leads to functional disorders in both digestibility and immunity, resulting in increased risks of eating disorders and inflammatory responses [12,13]. Besides being affected by host factors, the gut microbiota is sensitive to environmental stressors, reflected in the altered structure, composition, and functionality [14]. Benefitting from the rapid development of next-generation sequencing in the past decade, studies on microbial communities’ composition, diversity, and predicted function in different body parts have increased our understanding of microbial dynamics in both humans and animals [11].

Unlike ruminants, monogastric animals rely mainly on commensal bacteria in the gut for dietary fiber fermentation to produce short-chain fatty acids [15]. Since they profoundly influence host performance and health, the gut microbiota has received increased research attention in monogastric farm animals in the past decade. The diversity in the gut microbial community among individuals or populations results from combined environmental and host factors [16]. Recently, accumulating studies have highlighted the impact of high ambient temperatures on gut microbial diversities, composition, and biological functions in monogastric animal species, particularly on pigs and chickens [17,18,19].

This review aims to gather relevant current knowledge regarding the influences of heat stress on gut microbiota in monogastric animal species and discuss the use of gut microbiota composition as a potential biomarker of heat stress. The present paper is composed of three main sections following the introduction (Section 1). Section 2 reviews the intestinal damage and dysfunctions induced by heat stress. Section 3 summarizes the recent findings of primary responses in the gut microbiota to elevated ambient temperatures and discusses the use of gut microbiota composition as a potential biomarker of heat stress. Figure 1 provides a general overview of the main topics addressed in Section 2 and Section 3. Section 4 provides an overview of the current knowledge regarding strategies for heat stress alleviation through the mediation of gut microbiota. Lastly, the conclusions and future directions related to gut microbial research in monogastric agricultural animals are provided in Section 5.

## 2. Heat-Stress Induced Intestinal Damage and Dysfunctions

### 2.1. Intestinal Barrier Integrity

The small (duodenum, jejunum, and ileum) and large (cecum, colon, and rectum) intestines are a series of muscular tubes responsible for nutrient digestion and absorption, water and salt balance, and immune functions in the digestive system. The intestinal barrier on the luminal surface, made up of different types of epithelial cells covered by mucosal layers, together with tight junctions and immune cells, is critical in controlling the substance exchange between the gut tract and circulation system [20]. Additionally, this barrier system can produce diverse antimicrobial and immune molecules against pathogenic invasion [20]. The maintenance of the intestinal integrity by intercellular tight junctions and mucus is the foundation for a normal and stable intestinal system, which directly decides the overall homeostasis and health conditions [21]. Permeability is often used to evaluate intestinal integrity by measuring the flux rate of substances or molecules across the intestinal wall [22]. The intestinal integrity loss due to diseases or stress would result in increased permeability, which has been found to be associated with dysbiosis in gut microbiota [23,24].

### 2.2. Intestinal Integrity Loss and Dysfunctions under Heat Stress

Heat stress can disrupt the intestinal barrier, resulting in morphological damage and dysfunctions. Image analysis on intestinal histology is a common way to measure morphological indicators for loss in intestinal integrity [25]. Villus height, crypt depth, mucosal thickness, and villus height to crypt depth ratio are typical features generally measured from intestinal tissues for integrity evaluation [17,25,26]. An approximate 20% decrease in villus height measures of animals under heat stress has been reported across porcine and poultry studies [17,27]. Desquamation of the villi with even exposure of the lamina propria in some parts of the mucosal barrier in stressed laying hens was observed under the microscope [28]. Variations in crypt depth responding to heat exposure are noticed. No changes were found in crypt depth of either jejunum, ileum, and cecum in ducks exposed to heat stress (32 °C) for three weeks [25]. Increased crypt depth values were found in broilers exposed to the same high temperatures for two weeks and in growing pigs on the first three days under 35 °C heat stress [17,27]. However, the crypt depth then decreased on the seventh day in heat-stressed pigs compared with individuals in the thermoneutral group [27]. The total mucosal thickness decreased in rats under heat stress compared to the control [26]. To be noticed, different sections along the gut tract exhibited different responses to heat stress in terms of those indicators mentioned above in birds [17,25].

The morphological damage induced by heat stress has been shown to contribute to the increased permeability of the intestinal barrier, which allows luminal contents, such as lipopolysaccharide (LPS) and bacterial endotoxins, to enter the bloodstream and therefore causes inflammation and illness [9,29]. Reduced blood flow due to increased core body temperature under heat stress could be one reason driving the alternations in permeability [30]. Higher endotoxins levels were detected in the serum of broilers subjected to acute and chronic heat stress, indirectly indicating the increased permeability to bacterial products in the gut [31].

### 2.3. Relationship between Intestinal Integrity and Gut Microbial Community

Evidence indicates that the gut microbiota has a bidirectional relationship with the intestinal system. The gut microbiota and its products play as critical regulators for maintaining the intestinal barrier’s functions and integrity [15,32]. Short-chain fatty acids (SCFAs)—such as acetate, propionate, and butyrate—are the major products from bacterial fermentation of dietary fibers and play key functions in the host immune system by promoting the immune cell development and differentiation and increasing the secretion of anti-inflammatory cytokines [33]. Significantly, these SCFAs can optimize the growth and development of commensal bacterial species and help promote intestinal integrity by acting on tight junctions in the gut barrier [15,34]. Dysbiosis in gut microbiota induced by heat stress leads to a reduction in SCFAs production, contributing to the losses of intestinal integrity. The increased intestinal permeability of people under multiple physiological stressors has been found to be associated with changes in gut microbial composition and metabolism [24].

Moreover, a recent study based on longitudinal measures revealed a concomitant relationship between dysbiosis in gut microbiota and increased intestinal permeability [23]. Optimizing the gut microbiota may help protect the intestinal barrier’s integrity from heat stress [35]. The gut microbiota could play a key role in developing strategies to alleviate heat stress for farm animals.

## 3. Influences of Heat Stress on Gut Microbiota Structure, Composition, and Functionality

### 3.1. Core Gut Microbes in Swine and Chicken Population

Trillions of microbes belonging to more than 500 bacterial species are constantly present in mammals’ gut [36]. In vertebrates, up to 90% of the gut microbial community comprises two phyla: *Firmicutes* and *Bacteroidetes* [37,38]. The gut microbial community is dynamic and varies while aging [38,39]. The GI tract of newborn humans and animals is germ-free before subjected to maternal and external environments [36,40]. Petri and his colleagues documented the microbial succession process in pigs. They reported that *Clostridiaceae*, *Lactobacillaceae*, and *Streptococcaceae* families in *Firmicutes* phylum and *Enterobacteriaceae* family in *Proteobacteria* phylum were dominant in the gut of pre-weaned piglets [39]. Then, through a series of successive waves over time, the gut microbial community of pigs becomes mature and relatively stable at some point after weaning with a fixed diet and environment [41].

Similarly, several porcine studies have confirmed that *Firmicutes* and *Bacteroidetes* are the top two phyla—occupying more than 70% or even 90%—followed by *Proteobacteria* phylum during the post-weaning period, in the gut microbial community [19,42,43]. A recent study recorded the longitudinal dynamics of the gut microbiota throughout the lifetime in pigs and reported increased alpha diversity and species richness along the growth [43]. They also found several core microbes consistently present in the gut across growth stages [43]. Additionally, there are differences in the alpha-diversity of microbiota sampled from different gut parts and feces [42].

Similarly, the microbiome develops rapidly in the gut after hatching in chickens. Initially, the gut microbiota is dominated by *Enterobacteriaceae* belonging to *Proteobacteria* phylum; then it switches to *Clostridiales* from *Firmicutes* phylum rapidly within the first week after hatching [44]. Then, phyla *Firmicutes*, *Bacteroidetes*, *Proteobacteria*, and *Actinobacteria* become the main components in gut microbiota during the following growth period in broilers and laying hens [45,46]. However, the microbial profile varies across different gut parts (duodenum, jejunum, ileum, cecum, and colon), ages, and feeding patterns [46,47]. Compared to other GI tract parts, the cecum in a chicken has the most complex microbial community with greater richness and diversity [48,49]. Therefore, most studies targeted the microbiome collected from cecum when investigating the changes induced by heat stress in chickens. Similarly, because the gut microbiota composition changes across life stages, heat stress may have different effects on the gut microbiota of animals at varying ages [38,39]. Furthermore, it is well recognized that the diet significantly impacts the gut microbiota [42,50]. Given that many factors can shape the gut microbial community, it is critical to develop a solid experimental design that accounts for confounding variables, obtains sufficient sample size, and precisely quantifies the baseline or reference microbiota for comparison.

### 3.2. General Trends in Gut Microbial Structure Responding to Heat Stress

Animals on the farm or in transportation are exposed to unfavorable climatic conditions. Several studies investigating the birds’ physiological activities and productivity under different thermal conditions suggested that the thermal comfort zone is 18–28 °C for broilers and 25.9–29.9 °C for laying hens with relative humidity of 40–70% [51,52]. Within the range of thermal comfort zone, chickens are expected to achieve maximum productivity. When the air temperature exceeds the thermoneutral range, birds cannot release the extra metabolic heat from the body, which results in increased body temperature and heat stress [52]. Effects of chronic and acute heat stress ranging from 30 to 38 °C on the gut microbial community have been investigated to mimic various situations faced by animals during production and transportation. Most modern broilers can reach the market weight (4–5 pounds) during around six to eight weeks, while layers start laying eggs from 17 to 18 weeks of age and can be kept on farms for about 100 weeks. Thus, the exposure length to high ambient temperatures is different for broilers and laying hens under the practical production environment. Studies investigating the effects of heat stress generally applied two- to four-week thermal treatment for broilers [17,45,53], and more prolonged treatment for laying hens [18,54,55].

The gut microbiota of broilers and laying hens seems to respond to chronic cyclic heat stress differently. In broilers, heat stress tends to increase the species richness and diversity of either cecal or ileal microbiota, with an exposure time ranging from two to four weeks [17,45,53]. While in laying hens, Wang et al. reported a decreased species richness in hens after being exposed to heat stress for 10 weeks and the potential effects of mild cyclic heat stress on limiting the microbial growth in the cecum [54]. However, no difference in alpha diversity was noticed in the cecal microbiota of hens exposed to heat stress than the control group in another study with four-week exposure to heat stress [55]. The *Firmicutes-to-Bacteroidetes* ratio was significantly increased with increased abundance of *Firmicutes* and decreased abundance of *Bacteroidetes* observed in the gut microbiota of stressed broilers in all three broiler studies [17,45,53]. A similar pattern of the changes in *Firmicutes* and *Bacteroidetes* abundances was observed in laying hens exposed to heat stress for four weeks [55].

Interestingly, when the heat-stress exposure length was prolonged to 7–10 weeks, opposite results of the *Firmicutes-to-Bacteroidetes* ratio were noticed in two laying hen studies [18,54]. Consistently, distinct clusters of gut microbes were detected between chickens in thermoneutral conditions and those under heat stress across studies [17,45,53,54,55]. This difference in beta diversity reflects the variation in microbial compositions between thermal treatment and control groups.

The thermoneutral temperature is generally maintained around 18 to 21 °C to maximize growing pigs’ productivity. When the ambient temperature exceeds the range mentioned above, pigs are likely suffering from heat stress indicated by a series of changes in several physiological activities, such as increased respiration rate and rectal temperature and altered feeding patterns with reduced feed intake and elevated water consumption [56]. Increased respiration rate and decreased feed intake were observed in growing pigs when the ambient temperatures raised above 21.3 °C and 22.9 °C, respectively [57]. Unlike chicken studies, fecal samples taken from the rectum are generally used for 16S rRNA gene sequencing in swine studies to characterize the gut microbiota because they are easy to be collected without euthanizing the animal. Moreover, repeated measures of gut microbial data from the same individuals under different treatment conditions can be robust in statistical analyses with low variability.

In growing pigs, short exposure to heat stress can induce dysbiosis in the gut microbial community [58]. Under the thermoneutral condition, *Firmicutes* and *Bacteroidetes* are the top and the second abundant phylum, followed by *Proteobacteria* phylum at the third place, in the gut microbiota [19,58]. One-day acute heat stress has been found to reduce the gut microbial diversity and induce changes in composition, with *Proteobacteria* replacing *Bacteroidetes* as the second dominant phylum [58]. Besides, gut microbes were separated into different clusters between control and heat stress groups [58]. Greater bacteria richness was found in pigs exposed to mild heat stress (29 °C) for three weeks, while no difference was found in diversity than in the thermoneutral group [19]. Extending the heat stress to 13 weeks, less diverse gut microbiota was noticed and the boost in the relative abundance of *Firmicutes*, *Proteobacteria*, and *Spirochaetes* phyla [19]. However, gut microbiota exhibits resistance to heat stress in sows during late gestation [59]. Diversity and richness of gut microbiota were relatively stable in pregnant sows under heat stress [59]. At the same time, for composition at the phylum level, only *Spirochaetes* decreased in the relative abundance among several dominant phyla [59].

### 3.3. Gut Microbes Responding to Heat Stress Could Be Considered as Biomarkers

Identification of animals suffering from heat stress can serve as an essential strategy for precision livestock farming. The use of biomarkers for diagnosing diseases, monitoring the physiological state, and forecasting the risk of disease or low productive performance, has become a valuable tool assisting agricultural animal production [60]. Single or combination of molecular, physiological, and behavioral biomarkers are currently used for identifying heat-stressed animals or evaluate individuals’ heat resilience. To maintain normal homeostasis, animals try to adjust their behaviors initially to adapt to temperature changes. Feed intake, water intake, feeding, excretion frequency, or standing and lying are often monitored and considered behavioral indicators for early heat stress detection in livestock [61]. In poultry, heat stress increases the frequency and time spent in panting, wing stretching, sleeping, and sitting, but decreases feeding, standing, and walking activities [62]. Various proteins, inflammatory-related cytokines, blood cells, and bacteria products circulating with the bloodstream can also be used as biomarkers of heat stress. For instance, the serum heat shock protein 70 and cortisol in serum significantly increase in heat-stressed chickens [53,63]. Interleukin-10 (IL-10) is an important cytokine that suppresses inflammation, and it has been found associated with heat tolerance [64].

Additionally, considering blood cells, the heterophil-to-lymphocyte ratio developed in 1983 is a well-established criterion for measuring the stress and welfare status of the animal in numerous studies [65,66,67]. Moreover, the dysbiosis of commensal gut bacteria induced by heat stress releases endotoxin into the systemic circulation through the damaged intestinal mucosa and causes inflammatory responses [68]. Nonetheless, because of the complexity and multi-dimension of biological systems and physiological activities, there is no gold standard for few reliable biomarkers. Thus, the discovery of new biomarkers that can better identify heat-stressed or tolerant animals is needed.

The use of the gut microbial flora as biomarkers assisting in early disease diagnosis has been investigated in human studies [69,70,71]. These studies compared the gut microbiota between patients and healthy cases and also identified a set of differential bacteria profoundly associated with the disease under various conditions. A comprehensive summary of the gut microbes from many well-designed studies is always the initial step in developing a new biomarker. A series of gut microbial taxa has been identified with thermal stress through a case-control experimental design in pigs and chickens (Table 1). At the phylum level, fluctuations in the relative abundance of two predominant phyla *Firmicutes* and *Bacteroidetes* in the gut microbiota of stressed animals have been highlighted [17,18,19,53]. In rodent and human research, the *Firmicutes-to-Bacteroidetes* ratio has received significant attention recently and is widely considered an essential indicator for intestinal dysbiosis and inflammation [72]. It is commonly accepted that this ratio is associated with body mass index, weight gain, and obesity in humans [69,73]. In rodents and pigs, the ratio tends to increase in obese subjects compared to lean cases [69,74,75]. Increased *Firmicutes-to-Bacteroidetes* ratios were observed mainly in broiler and swine studies where animals were exposed to acute heat stress less than four weeks [17,19,45,53,58]. However, in laying hens, no changes or a decline in the ratio was addressed when animals were subjected to prolonged heat stress for more than four weeks [18,54,55]. Thus, the exposure length to high ambient temperatures could be a possible variable to consider when evaluating gut microbiota composition as a biomarker. For example, some individuals may develop adaptations or tolerance to cyclic chronic thermal loads [7].

Several genera—including *Acinetobacter*, *Bacteroides*, *Coprococcus*, *Dorea*, *Faecalibacterium*, *Lactobacillus*, and *Streptococcus*—were identified as responding to heat stress in multiple studies (Table 1). These microbial taxa may have a significant impact on host growth, feed efficiency, and health. The genus *Acinetobacter* was positively associated with abdominal fat content in ducks [25]. The beneficial effects of genus *Bacteroides* on host intestinal health and the relationship with obesity have been widely reviewed [76]. Many studies have observed that *Coprococcus*, *Dorea*, and *Faecalibacterium* were highly abundant in animals with good feed efficiency and productive performance [77,78,79,80]. *Lactobacillus* strains are commonly used as probiotics and can benefit both humans and animals by preventing intestinal disorders and increasing overall gut health [81,82]. Furthermore, He et al. demonstrated that *Lactobacillus* might help ducks with antioxidant activity under heat stress [25]. The genus *Streptococcus* was also positively related to pig growth during the nursery stage [43,59].

Differential microbial taxa identified between heat stress and thermoneutral groups could be considered as biomarker candidates of heat stress. However, changes in the gut microbial composition may be related to reduced feed intake induced by heat stress [55,58]. Thus, including a pair-feeding group to account for feed intake in the study helps identify microbes that respond directly to high temperatures [55,58]. The discovery of new biomarkers is a complex process that requires accurate measures of causal associations between the gut microbiota and thermal conditions, validation of the findings in different populations under diverse conditions, and a comprehensive understanding of the biomarker’s mechanism. Therefore, given that we are in the initial stages of investigating the effects of heat stress on the gut microbiota in agricultural animals, we expect the number of studies looking at this topic to increase substantially in the next few years.

### 3.4. Associated Changes in Metabolic Functions and Metabolites of Differential Gut Microbes

Changes in the microbial composition induced by high temperatures result in altered metabolic functions and associated metabolites of the gut microbial community. The production of SCFAs from bacterial fermentation is important evidence that the gut microbiota participates in host carbohydrate metabolism and energy homeostasis in the cecum and colon [83]. Three main SCFAs, including butyrate, acetate, and propionate, are known as primary energy sources for various intestinal cells and other commensal bacteria in the gut [83]. Butyrate supports intestinal barrier integrity and immune functions by enhancing the assembly of tight junctions and promoting the differentiation of regulatory T cells [33,34]. Acetate produced by the gut microbiota into plasma serves as an essential energy source for the muscular system during exercise or fasting [84]. Similarly, propionate regulates satiety and reduces the risk of diet-induced obesity due to its critical role in promoting insulin sensitivity and enhancing gluconeogenesis in the liver [85,86].

An acute heat-stress treatment was found to reduce the levels of fecal butyrate, acetate, and propionate in growing pigs [58]. Similarly, heat stress was found to lower the concentrations of propionate, butyrate, and total SCFAs, while it had no effects on the acetate concentration in feces of pregnant sows [59]. The reduction of fecal SCFAs has been associated with changes in the abundance of several SCFAs-producing bacteria induced by heat stress, like *Clostridiales* belonging to *Firmicutes* phylum and *Bacteroidales* from *Bacteroidetes* phylum [58,59]. Moreover, the gut microbiota exhibits prominent influences in amino acid metabolism because of its proteolytic capacity in protein degradation and amino-acid utilization [87]. Amino acids (aspartate and β-alanine) and several organic acids (malate, lactate, and fumarate) were measured at a lower level in pregnant sows subjected to heat stress compared to the control ones and were found associated with several predominant taxa in the gut microbiota [59]. Limited information is available about the microbial metabolite profile of birds under heat stress. More studies are needed to further understand the interplay between gut microbiota, metabolites, and their functions on host physiology under high temperatures.

Functional analysis of the gut metagenomes allows us to better understand the functional capacity of the gut microbiota and identify potential gene markers related to bacterial metabolites under different environmental conditions [88,89]. Functional annotation of bacterial gene sequences against the Clusters of Orthologous Groups of proteins (COGs) or Kyoto Encyclopedia of Genes and Genomes (KEGG) databases is a widely used approach to characterize the metabolic functions of the gut microbial community. In laying hens under thermoneutral conditions, several COG functional categories—such as replication, recombination, and repair, carbohydrate and amino acid transport, and metabolism—were notably associated with the fecal microbiota [18]. For KEGG functional pathway, most microbial genes were mapped to functions related to the metabolism of carbohydrates, amino acids, nucleotide, and energy in the same study [18]. Tian et al. identified transcription, amino acid metabolism, and carbohydrate transport and metabolism as the top three predicted functions of the gut microbial community in ducks under thermoneutral conditions [90]. Several studies revealed the changes in metabolic functions of gut microbiota of birds under heat stress. Zhu et al. indicated that high temperatures increased the enrichment of benzoate degradation and cysteine and methionine metabolism pathways but decreased the retinol metabolism and phenylpropanoid biosynthesis pathways in laying hens [18]. While in ducks, heat stress was found to enrich starch and sucrose metabolism, transcription machinery, energy metabolism, and methane metabolism pathways but lower ABC transporters and several signaling pathways [90]. Functional changes in the gut microbiota induced by high temperatures provide more information to understand the metabolic role of the gut microbiome in host health.

## 4. Strategies to Alleviate Heat Stress through Mediating Gut Microbiota

### 4.1. Administration of Probiotics and Prebiotics in Regulating Gut Microbiota

Due to its importance in host health and nutrition, the gut microbiota has been targeted for developing therapies or diagnostic approaches for diseases and disorders in numerous studies in humans. For food animal production purposes, modulating the gut microbiota has recently become a favored objective to improve performance and productivity, enhance welfare, and moderate the detrimental effects of various environments and diseases. Probiotics and prebiotics have been investigated for decades for their beneficial effects on intestinal functions and disorders. Probiotics are a group of live bacteria or yeasts, such as *Lactobacilli* and *Bifidobacterial*, used as a food ingredient to help balance the microbial community, promote gut health, and modulate digestive activities [91,92]. Differently, prebiotics is a food compound that cannot be digested but can help keep the host healthy by promoting the growth of beneficial microbes and shaping microbial composition in the gut [93].

Dietary supplementation of probiotics or prebiotics has been considered a promising management tool to improve agricultural animals’ overall performance and health [94,95]. Numerous studies in chickens have confirmed the beneficial and protective effects of probiotics and prebiotics supplementations on restoring microbial homeostasis by regulating the composition and structure of the gut microbiota to prevent heat stress-induced complications. Early in 2004, Lan et al. found that the food-supplied probiotics helped keep the stability of gut microbiota in response to heat stress in broilers [82]. Later, Song et al. reported increased levels of lactic acid-producing bacteria (*Lactobacillus* and *Bifidobacterium*) and decreased counts of pathogenic bacteria (*coliforms* and *Clostridium*) in heat-stressed broilers supplied with the probiotic mixture (*Bacillus licheniformis*, *Bacillus subtilis*, and *Lactobacillus plantarum*) compared to the control group [96]. Correspondingly, Zhang et al. studied the effects of probiotics mixture (*Bacillus subtilis* and *Enterococcus faecium*) on regulating the gut microbial composition in heat-stressed laying hens and found that the probiotics treatment optimized the microbial structure by reducing the growth of pathogenic bacteria *E. coli* while elevating the amount of beneficial bacteria *Lactobacillus* [28].

Additionally, probiotics supplement helped recover the reductions in egg production rate, average daily feed intake, and average egg weight caused by heat stress [28]. The genus *Lactobacillus*, a gram-positive bacteria that can be found in the GI tract of humans and mammals, is a significant member of probiotics and can be widely applied in food fermentation [97]. Its beneficial effects on host health have been studied well, such as maintaining the gut microbial homeostasis, promoting the immune functions against intestinal inflammation, keeping intestinal barrier integrity, and protecting the host from pathogen invasions [81]. Sohail et al. investigated the effects of prebiotics, probiotics, and their mixture on the gut microbiota and observed a higher species richness of the microbial community in heat-stressed broilers supplied with prebiotics and mixture compared to heat-stressed individuals in the control group [98]. Additionally, the prebiotics treatment (*Saccharomyces cerevisiae*) helped keep the intestinal barrier integrity and microbial balance by enhancing the presence of beneficial bacteria in rats’ gut in responses to heat stress [35]. Most beneficial bacteria are significant SCFAs producers and essential players in antioxidant and anti-inflammation activities [35,99].

Several types of probiotics and prebiotics, such as lactic acid-producing bacteria, yeast, and *Bacillus* species, have been suggested as dietary supplements to improve health and productivity in the pork industry [94]. Lv et al. estimated the effects of selenium-enriched probiotics on the gut microbiota of weaning piglets raised in high ambient temperatures for 42 days [100]. On the 28th and 42nd days of the experimental trial, the levels of *Lactobacillus* in the gut microbiota were found to be higher in groups supplied with probiotics than the control. Distinguished clusters of microbial composition were separated between treatment and control groups on the last day of the study. Compared to controls under heat stress, piglets given selenium-enriched probiotics had higher average daily gain and final body weight, as well as a reduced diarrhea rate [100]. The findings of this study highlighted the protective effects of probiotics on maintaining the stability of the gut microbiota and performance in piglets raised in high temperatures [100]. However, more studies are needed to determine the effectiveness of probiotics and prebiotics in enhancing heat tolerance by regulating pigs’ gut microbiota.

### 4.2. Other Potential Methods: The Selection of Enterotypes for Heat Resilience/Tolerance

Identifying heat-tolerant animals is still challenging because of the complicated physiological responses to elevated ambient temperatures in animals and difficulties related to selecting suitable biomarkers [101]. As more and more studies investigate the associated changes in different taxonomic ranks of the gut microbiota responding to high temperatures, individual taxa and collections of taxa could be considered as biomarkers to help select heat-tolerant animals. Most studies identified individual microbes or taxa responding to heat treatment by comparing the gut microbial compositions. A limited number of studies aimed at evaluating changes in the interactions between animals’ gut microbes under heat stress. Taxonomic correlation network analysis has been used in some studies to detect the interactions between microbial features associated with the factors of interest [43,102]. Cao et al. constructed two ecological networks in the gut microbiota of rats in heat-acclimation and control groups and reported variations in the robustness and degree distribution between two networks [102]. Through this network analysis, groups of microbes associated with heat tolerance or susceptibility can be identified as biomarkers. However, more microbiota transplant experiments are still needed to confirm the effects of gut microbiota in regulating host heat tolerance capacity.

Moreover, a growing body of research has indicated that host genotype directly affects gut microbial composition by identifying a portion of heritable gut microbiota [103,104]. Regions or specific loci on the genome were found to contribute to the variations in the gut microbial community [103,104]. Thus, the abundance or composition of the gut microbial community could be considered quantitative traits in genetics selection to improve an animal’s ability in heat acclimation.

## 5. Conclusions

The ongoing climate change has the potential to further exacerbated the negative impacts of heat stress on agricultural animal production. A growing number of studies have focused on developing strategies to alleviate heat stress and improve heat tolerance of agricultural animals in the past decade. The gut microbiome impacts host health and welfare through its crucial regulatory effects on host immune functions and metabolism. Dysbiosis in the gut microbial community may lead to the occurrence of inflammation, disorders, and even diseases. Heat stress has been proven to induce a series of modifications in the gut microbial structure, composition, and functionality in monogastric animal species, such as chickens and pigs. Current microbial studies mainly focus on documenting the changes in the gut microbiota and microbial metabolites and their relationships with other physiological responses under different thermal conditions. These findings enable us to better understand the importance of gut microbiota in host health and highlight the value of the gut microbiota composition as a potential biomarker of heat stress. Different lengths of heat exposure may induce diverse reactions in the gut microbiota. Future studies are still needed to clarify this question and validate the findings in different populations, especially in swine. Most studies currently rely on the 16S rRNA sequencing technology that focuses on specific genetic regions due to its affordable price. As the sequencing becomes cheaper, the use of shotgun metagenomic sequencing, which covers the entire DNA content, will provide a more comprehensive understanding of gut microbiota in related research. Additionally, the development of consistent approaches to collect samples and analyze data is helpful to compare and summarize the results from different studies about how the gut microbiota responds to heat stress in multivariate space.

## Figures and Tables

**Figure 1 animals-11-01833-f001:**
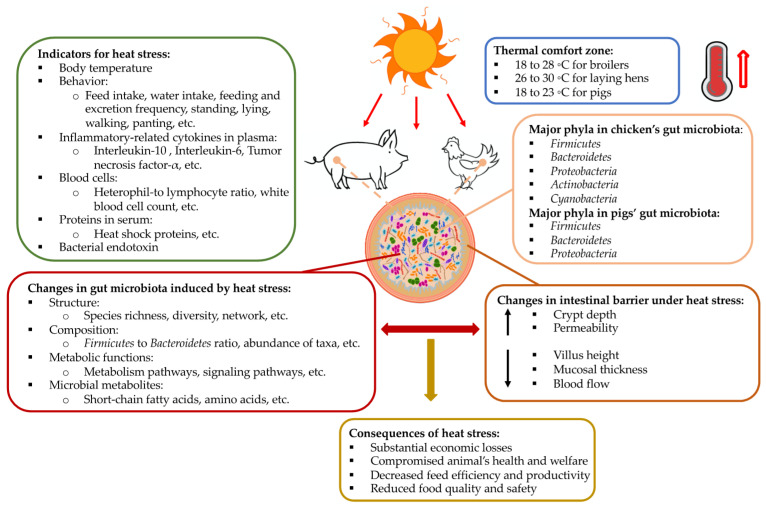
A schematic summary of the impact of high ambient temperatures on the gut microbiota and intestinal morphology in monogastric agricultural animals: pig and chicken. Other related information—including commonly used biomarkers of heat stress, thermoneutral zones for pigs and chickens, main components of the gut microbiota, and consequences of heat stress—are also included.

**Table 1 animals-11-01833-t001:** Summary of published alternations in the gut microbiota induced by heat stress.

Animal Species	Start Age	Sample Source	Treatment ^a^	Changes in Phylum (HS vs. TC)	Changes in Genus (HS vs. TC) ^b^	References
Broiler	3 weeks	Cecum	HS *: 31–33 °C TC *: 21–23 °C 2 weeks	*Firmicutes* (+) *Bacteroidetes* (-)	*Anaerofustis* (+) *Pseudonocardia* (+) *Rikenella* (+) *Tyzzerella* (+) *Parabacteroides* (-) *Romboutsia* (-) *Saccharimonas* (-) *Weissella* (-)	[17]
Broiler	4 weeks	Ileum	HS *: 30–32 °C TC *: 20–22 °C 2 weeks	*Firmicutes* (+) *Bacteroidetes* (-) *Proteobacteria* (-)	*Alistipes* (+) *Azospirillum* (+) *Clostridium XIVb* (+) *Faecalibacterium* (+) *Oscillibacter* (+) *Rothia* (+) *Streptophyta* (+) *Coprococcus* (-) *Streptococcus* (-)	[45]
Broiler	2 weeks	Cecum	HS *: 34–38 °C TC *: 24–26 °C 4 weeks	*Firmicutes* (+) *Proteobacteria* (+) *Bacteroidetes* (-) *Cyanobacteria* (-)	*Lactobacillus* (+) *Bacteroides* (-) *Dorea* (-) *Faecalibacterium* (-) *Oscillospira* (-)	[53]
Laying hen	11 weeks	Cecum	HS *: 24–30 °C TC *: 22 °C 10 weeks	*Bacteroidetes* (+) *Euryarchaeota* (-)	*Akkermansia* (+) *Desulfovibrio* (+) *Faecalibacterium* (-)	[54]
Laying hen	45 weeks	Feces	HS *: 25–34 °C TC *: 21–28 °C 6.6 weeks	*Bacteroidetes* (+) *Firmicutes* (-) *Fusobacteria* (-) *Proteobacteria* (-)	*norank_p_Candidatus_Saccharimonas* (+) *Cloacibacillus* (+) *Enterorhabdus* (+) *Flavihumibacter* (+) *Prolixibacter* (+) *Sunxiuqinia* (+) *norank_f_Synergistaceae* (+) *Synergistes* (+) *Dorea* (-) *Exiguobacterium* (-) *norank_f_Lachnospiraceae* (-) *Marvinbryantia* (-)	[18]
Laying hen	28 weeks	Cecum	HS *: 29–35 °C TC *: 20–22 °C PF: 20–22 °C 4 weeks	No differences	*Anaerosporobacter* (+) *Barnesiella* (+) *Clostridium_sensu_stricto_1* (+) *Escherichia_Shigella* (+) *Gallibacterium* (+) *Odoribacter* (+) *Rikenellaceae_RC9_gut_group* (+) *Sphaerochaeta* (+) *Bacteroides* (-) *Lactobacillus* (-) *Prevotellaceae_Ga6A1_group* (-) *Ruminococcaceae_UCG_005* (-)	[55]
Duck	5 weeks	Jejunum Ileum Cecum	HS *: 32 °C TC *: 25 °C 3 weeks	*Proteobacteria* (+) (Jejunum, Cecum) *Firmicutes* (-) (Jejunum, Cecum) No differences in ileum	*Acinetobacter* (+) (Jejunum) *Mitochondria* (+) (Jejunum) *Lactobacillus* (-) (Jejunum) No sig difference in ileum	[25]
Pig	11 weeks	Rectum	HS *: 26 °C TC *: 25 °C 12 weeks TC * to HS *: 28–30 °C 3 weeks	*Firmicutes* (+) *Proteobacteria* (+) *Spirochaetes* (+) *Actinobacteria* (-) *Bacteroidetes* (-)	Not reported	[19]
Pig	Not mentioned (29–31 kg body weight)	Rectum	HS *: 34–36 °C TC *: 24–26 °C PF: 24–26 °C 24 h	*Proteobacteria* (+) *Bacteroidetes* (-)	*Acinetobacter* (+)	[58]
Pig	First-parity (85 d in gestation)	Feces	HS *: 28–32 °C TC *: 18–22 °C 4 weeks	No differences	*Coprococcus 3* (+) *Coprostanoli-genes group* (+) *Halomonas* (+) *Ruminococcaceae UCG-005* (+) *Ruminococcaceae UCG-013* (+) *Bacteroidales RF16 group* (-) *Streptococcus* (-) *Treponema 2* (-)	[59]

^a^ HS = heat stress group; TC = thermoneutral control group; PF = pair-feeding group on the daily feed intake of HS group. ^b^ The plus sign (+) indicates the increase in the abundance of the given microbe of animals under heat stress compared to the thermoneutral control group. The minus sign (-) indicates the decrease in the abundance of the given microbe of animals under heat stress compared to the thermoneutral control group. * Animals in the group had ad libitum access to feed and water.

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
