# Peer review of "Potential Use of Gut Microbiota Composition as a Biomarker of Heat Stress in Monogastric Species: A Review"

_animals, 2021, doi:10.3390/ani11061833_

Round 1

Reviewer 1 Report

I understand that the authors are trying to connect how global warming could impact animal productivity; however, wording needs to be changed as it appears that the authors are stating that because of global warming, we have heat stress.  Heat stress occurs any time there is a rapid increase in temperature and humidity, which can occur during the changing of seasons or summer months and is not wholly tied to global warming.   This needs to be adjusted throughout the paper.

Remove introductory sentence in the simple summary at line 7.  Heat stress is not a new phenomenon associated with climate change.  

Reword line 16.  Heat stress could become an increasing issue if climate temperatures continue to rise.  

Where are citations for sentence in line 17?  If no citations, do not use this sentence

Line 52, remove the "due to global warming".  Heat stress happens when seasons change.  

Lines 434-439.  Remove.  These are redundant and the sentence 434 is not accurate.

Line 463, remove the words "to worsen".  

A section needs to be added to discuss why the phyla and genera that are discussed in this paper are important in terms of animal growth, efficiency and health beyond SCFAs.

Author Response

I understand that the authors are trying to connect how global warming could impact animal productivity; however, wording needs to be changed as it appears that the authors are stating that because of global warming, we have heat stress. Heat stress occurs any time there is a rapid increase in temperature and humidity, which can occur during the changing of seasons or summer months and is not wholly tied to global warming. This needs to be adjusted throughout the paper.

Author’s response: We appreciate the reviewer for pointing this out, and we agree with the reviewer about the relationship between heat stress and global warming. We have carefully checked every sentence and made adjustments to eliminate the misleading throughout the manuscript. First three paragraphs in the Introduction have been restructured to correct the relationship between heat stress and global warming. Also, revised sentences have been highlighted and listed in terms of line and page numbers under each comment.

Remove introductory sentence in the simple summary at line 7.  Heat stress is not a new phenomenon associated with climate change.  

Author’s response: We have removed the sentence indicated by the reviewer and revised the first sentence in the Simple Summary accordingly. The revised sentence can be found in Line 7-8 on Page 1.

Reword line 16.  Heat stress could become an increasing issue if climate temperatures continue to rise.  

Author’s response: We thank the reviewer for providing such a good sentence. We have rewritten the sentence suggested by the reviewer. The new sentence can be found in Line 15-16 on Page 1 in the Abstract.

Where are citations for sentence in line 17?  If no citations, do not use this sentence

Author’s response: We have rewritten the sentence to avoid the concern about citation. It can be found in Line 16-18 on Page 1.

Line 52, remove the "due to global warming".  Heat stress happens when seasons change.  

Author’s response: We have reconstructed the whole paragraph to correct the relationship between heat stress and global warming. This sentence has been removed in the revised paragraph.

Lines 434-439.  Remove.  These are redundant and the sentence 434 is not accurate.

Author’s response: We appreciate the reviewer’s insightful suggestion! We have removed the sentences.

Line 463, remove the words "to worsen".  

Author’s response: We have removed the words suggested by the reviewer and also reworded the sentence. The new sentence can be found in Line 472-473 on Page 11.

A section needs to be added to discuss why the phyla and genera that are discussed in this paper are important in terms of animal growth, efficiency and health beyond SCFAs.

Author’s response: We thank the reviewer for providing a great idea for us to improve our content. To make the content flow well, we have added the discussions suggested by the reviewer under section 3.3 for both phyla and genera. The discussions can be found in Line 297-301 and Line 309-321 on Page 7 for phyla and genera, respectively.

Reviewer 2 Report

Dear Authors 

Your review look like really interesting. I have several suggestions to be discussed: 

Section 3.2  You introduce the tropical countries concept , that is right , however today the rang of temperatures in side the poultry farm in mediterranean areas for instance reach temperature for above the thermal comfort.

Section 4.1  . It will be interesting to identify the most important indicators in order to identify animal welfare. In principal the most simple could be zootechnical parameters.

Author Response

Dear Authors 

Your review look like really interesting. I have several suggestions to be discussed: 

Section 3.2  You introduce the tropical countries concept , that is right , however today the rang of temperatures in side the poultry farm in mediterranean areas for instance reach temperature for above the thermal comfort.

Author’s response: We appreciate that the reviewer brought up an excellent point here. We agree that heat stress has currently impacted larger areas worldwide. We have rewritten the sentences to address the reviewer’s concern. They can be found in Line 199-201 on Page 5 at the beginning of section 3.2.

Section 4.1  . It will be interesting to identify the most important indicators in order to identify animal welfare. In principal the most simple could be zootechnical parameters.

Author’s response: Following the reviewer’s suggestion, we have added information about zootechnical parameters from poultry and swine studies in section 4.1. Zootechnical parameters, such as average daily gain, body weight, egg production rate, feed intake, diarrhea and mortality rate, are commonly used in studies investigating the effectiveness of probiotics and prebiotics under heat stress. The added sentences can be found in Line 418-419 on Page 10 and in Line 440-442 on Page 11, respectively.

Reviewer 3 Report

The paper is well written and provides basic information. The objective of the review is of the paper is clear. Some paragraphs could be shorter and more to the point. 

The review is a cumulation of observations and does not provide conceptual hypothesis based on these observations. the paper provides enough information to at least discuss factors affecting the microbiome and to give an opinion on the value of the microbiome as biomarker. 

The most important conclusion of the authors is described in line 201-204, although don't specify their  ïdeal"design.

The review would have been more valuable if the cascade of processes affecting the microbiome would have been discussed. I realize that it it complex, especially due to the fermentation products effect on the intestinal integrity, directly or via systemic mechanisms. Not only SCFA's but also endotoxins and biogenic amines play a role. The latter 2 have a negative effect on intestinal integrity. That makes figure 1 to simple. 

The processes eventually influence the amount and composition of the substrate available to the microflora. That effect differs in different parts of the GIT. This may explain the differences in the observations.

I have the opinion that the paper would be of more value if;

  1. The effects of short term and chronic heat stress would be discussed separately.
  2. In order to be able to determine the microbiome can be used as a biomarker for heat stress only experiments in which is corrected for feed intake, for example by pair feeding, can be conclusive. Most of the experiments mentioned in this paper did not correct for it. Xiong e.a. did correct and they did not observe large differences. the pigs were relatively  old. It is advised to mention in table 1 all treatments in the experiments.
  3. The difference in age could be mentioned separately. In both layers and sows the effect were smaller than in young animals. I suggest the authors to correlate the the effects with feeding level ( X* maintenance requirement) of the controls. 

Minor remarks:

line 20: what do you mean by energetic harvest

line 90-93: the sections do not correspond with the numbers of the sections. 

line 146 to 151 do not add to the paper. 

line 168: The statement suggests that there is a solution. Or is the microbiome a result of the integrity of the intestinal wall ?

Author Response

The paper is well written and provides basic information. The objective of the review is of the paper is clear. Some paragraphs could be shorter and more to the point. 

The review is a cumulation of observations and does not provide conceptual hypothesis based on these observations. the paper provides enough information to at least discuss factors affecting the microbiome and to give an opinion on the value of the microbiome as biomarker. 

The most important conclusion of the authors is described in line 201-204, although don't specify their  ïdeal"design.

Author’s response: We appreciate the time that the reviewer dedicated to reading and providing insightful comments to help us improve our manuscript. We have gone through the whole manuscript and removed sentences that seem redundant from multiple places. Especially, we restructured the first three paragraphs and removed redundant content to get to the point more directly. We reconstructed the sentence mentioned by the reviewer to better deliver our conclusion and also added some information to make it flow well. This update can be found in Line 191-197 on Page 5.

The review would have been more valuable if the cascade of processes affecting the microbiome would have been discussed. I realize that it it complex, especially due to the fermentation products effect on the intestinal integrity, directly or via systemic mechanisms. Not only SCFA's but also endotoxins and biogenic amines play a role. The latter 2 have a negative effect on intestinal integrity. That makes figure 1 to simple. 

The processes eventually influence the amount and composition of the substrate available to the microflora. That effect differs in different parts of the GIT. This may explain the differences in the observations.

Author’s response: We agree with the reviewer that it would be much better to describe the processes of how heat stress affects the gut microbiota. However, it is a very complex process that also includes many different pathways with many variables (mediators) playing different roles. It would be an excellent topic for a review paper in the future because it is worthy to include a large amount of information to describe the process in detail instead of briefly discussing this in the current manuscript (also in consideration of the limited amount of space available for this mini-review). The present review aimed at providing information on how the gut microbiota responds to heat stress and discuss its potential role as a biomarker of heat stress in swine and poultry. The processes and underlying mechanisms of effects of heat stress would be described in a major section, which may distract the focus of this review to some extent. Also, Figure 1 aims to present a big picture of elements or factors referred in this manuscript instead of depicting detailed relationships among each other.

I have the opinion that the paper would be of more value if;

  1. The effects of short term and chronic heat stress would be discussed separately.

Author’s response: We appreciate the reviewer’s valuable suggestions that help us improve the manuscript. We agree with the reviewer’s opinion, and we also intended to discuss the effects of acute and chronic heat stress separately in different sections at the beginning. However, we found that minimal research has been done to investigate the impact of heat treatment varying in length on the gut microbiota in swine. Also, the “short term” or “chronic heat stress” is a relative term, which depends on the production cycle for each animal species. Thus, in section 3.2, we intended to discuss the effects of heat stress separately by animal species. And within each species, we discussed the impacts of short-term heat stress and then chronic heat stress. With more studies conducted in the future, it is better to separately discuss the effects of heat stress by exposure length.

  1. In order to be able to determine the microbiome can be used as a biomarker for heat stress only experiments in which is corrected for feed intake, for example by pair feeding, can be conclusive. Most of the experiments mentioned in this paper did not correct for it. Xiong e.a. did correct and they did not observe large differences. the pigs were relatively old. It is advised to mention in table 1 all treatments in the experiments.

Author’s response: We thank the reviewer for providing such insightful comments. Following the reviewer’s advice, we have added information about all treatments, including pair-feeding group and access to feed and water for each study in Table 1, which can be found in Line 334-336 on Page 8-9. Two of the studies listed in Table 1 corrected for feed intake by including the pair-feeding group (Xing et al., 2019 for laying hens and Xiong et al., 2020 for pigs). Moreover, we have added several sentences to discuss this concern. The added discussion can be found in Line 322-326 on Page 7.

  1. The difference in age could be mentioned separately. In both layers and sows the effect were smaller than in young animals. I suggest the authors to correlate the the effects with feeding level ( X* maintenance requirement) of the controls. 

Author’s response:  Following the reviewer’s suggestion, we have added age and feeding level for each study listed in the Table 1 to provide more information. The updated Table 1 can be found in Line 334-336 on Page 8-9. Besides, we have added a discussion about considering differences in animal’s age when studying the effects of heat stress on gut microbiota, which can be found in Line 191-193 on Page 5.

Minor remarks:

line 20: what do you mean by energetic harvest

Author’s response: When we mentioned “energy harvest”, we intended to indicate that the gut microbiota can help the host’s digestion and absorption by producing metabolites and also can contribute energy to the host. We have changed the “harvest” to “balance” to eliminate the confusion. The updated phrases can be found in Line 19 on Page 1 and Line 62 on Page 2.

line 90-93: the sections do not correspond with the numbers of the sections. 

Author’s response: We appreciate that the reviewer pointed it out. We have fixed the numbers of the sections. The updated part can be found in Line 83-92 on Page 2.

line 146 to 151 do not add to the paper. 

Author’s response: We have removed this part from the manuscript suggested by the reviewer.

line 168: The statement suggests that there is a solution. Or is the microbiome a result of the integrity of the intestinal wall?

Author’s response: The statement suggests that modulation of gut microbiota may help maintain the integrity of the intestinal barrier under heat stress because the gut microbiota and its products have been found to play as critical regulators for maintaining the intestinal barrier’s functions and integrity. We have reworded the sentence to reduce the confusion. The sentence can be found in Line 157-158 on Page 4.

Round 2

Reviewer 1 Report

Please remove sentence on line 36.  It isn't clear as to what the stressors are or why you have identified only these two species.

Remove sentences 199-201.  Heat stress has occurred in other parts of the world for a long time not just the tropics.  Again, this sentence is misleading.  Heat stress is due to any rapid change in temperature.  This can occur anywhere at anytime when seasons change.

Line 472 - you have not proven this statement.  Instead of the word "has" you could use the words "has the potential to further increase the negative..."  Has is not the acceptable word here

Author Response

The sentence in line 36 was removed. 

That sentence in lines 199-201 was removed, the following sentence has been changed. Now lines 198-199.

The sentence in line 472 has been modified. Now line 469.

Reviewer 3 Report

The authors responded accurately to the remarks and suggestions. I realize that some of my remarks would be more suitable for an other review. 

I do not have further remarks

Author Response

AU: We appreciate the comments from the Reviewer, perhaps we will use them for the next review paper.